# Photoluminescence of Cesium-Doped Sodium Iodide Films Irradiated by UV LED

**DOI:** 10.3390/nano13202747

**Published:** 2023-10-11

**Authors:** Hsing-Yu Wu, Yu-Hung Kuan, Guoyu Yu, Yung-Shin Sun, Jin-Cherng Hsu

**Affiliations:** 1System Manufacturing Center, National Chung-Shan Institute of Science and Technology, New Taipei City 237209, Taiwan; andy810301@gmail.com; 2Center for Astronomical Physics and Engineering, Department of Optics and Photonics, National Central University, Taoyuan City 320317, Taiwan; 3Department of Electro-Optical Engineering, National Taipei University of Technology, Taipei 10608, Taiwan; 4Department of Physics, Fu Jen Catholic University, New Taipei City 242062, Taiwan; zmichael77@yahoo.com.tw; 5Department of Engineering and Technology, School of Computing and Engineering, University of Huddersfield, Queensgate, Huddersfield HD1 3DH, UK; g.yu@hud.ac.uk; 6Graduate Institute of Applied Science and Engineering, Fu Jen Catholic University, New Taipei City 242062, Taiwan

**Keywords:** scintillator, sodium iodide, Cs-doping, post-annealing, self-trapped exciton

## Abstract

Alkali metal halides have long been used as scintillators for applications as sensors and detectors. Usually, a small amount of impurities are added to these inorganic materials to improve their luminescence efficiencies. We investigate the structures and luminescent properties of un-doped sodium iodide (NaI) and cesium-doped NaI (NaI:Cs) films deposited by thermal vacuum evaporation. Instead of using the toxic element thallium (Tl), we introduced cesium dopant into NaI. This is the first study for the NaI:Cs film excited by UV LED’s ultraviolet C (273 nm, 4.54 eV). The luminescence spectra show two main peaks at 3.05 and 4.32/3.955 eV (for fused silica/B270 substrate), originating from the intrinsic defects and/or activator excited states and the intrinsic self-trapped excitons (STEs), respectively. In general, both Cs-doping and post-annealing processes enhance the luminescence performance of NaI films.

## 1. Introduction

A scintillator is a luminescent material that strongly absorbs ultraviolet (UV) light or ionizing radiation such as X-rays, gamma rays, and various particles [1,2,3,4,5]. After being excited by photons of high energy, its electrons in the valence band (VB) jump over the band gap to the conduction band (CB), leading to the subsequent scintillation of light [6]. As its excited states are unstable, the relaxation process may be delayed from a few nanoseconds to even hours, depending on the material [7,8]. Due to their photoluminescence (PL) properties, scintillators are commonly used as sensors in charge-coupled devices (CCD) or complementary metal-oxide-semiconductor (CMOS) detectors [9,10]. They are also applicable to medical diagnostics such as radiography, computer tomography (CT), positron emission tomography (PET), gamma imaging, and dosimetry [11,12,13,14]. Recent advancements and increasing interests in aerospace and remote sensing enable the use of scintillators as detectors in a variety of space missions such as the HERMES-TP/SP (High Energy Rapid Modular Ensemble of Satellites-Technological and Scientific Pathfinder) mission, the SPORT (Scintillation Prediction Observations Research Task), and the ALTEA (Anomalous Long Term Effects on Astronauts) project [15,16,17].

The three commonly used scintillator types are gas scintillators, organic crystals, and inorganic crystals. Gaseous scintillators consisting of nitrogen (N_2_) and low-pressure noble gases such as argon (Ar), helium (He), and xenon (Xe) are applied to the detection of heavily charged particles [18]. Typical disadvantages of these gas-based detectors include low sensitivity to small-sized chambers and susceptibility to humidity and barometric pressure changes. Organic scintillators, mainly composed of elements of low atomic numbers such as hydrogen (H), carbon (C), nitrogen (N), and oxygen (O), can be made in liquid or plastic forms [19,20,21]. Due to low interaction probability, their applications in detecting X-rays and gamma rays are limited [19]. Inorganic scintillators, usually crystals grown under high-temperature environments, can distinguish between gamma rays and neutrons/alpha particles [22]. These inorganic compounds can be classified into intrinsic, involving direct electron-hole recombination, and extrinsic, associated with impurities and additive dopants [23]. Among these materials, alkali metal halides are the most widely used compounds because of their efficient scintillation [24,25]. Examples include sodium iodide (NaI), cesium iodide (CsI), cesium fluoride (CsF), potassium iodide (KI), and lithium iodide (LiI). Usually, a small amount of impurities are added to the alkali metal halide to generate an energy state in the forbidden gap for converting X-rays and gamma-rays into visible (VIS) light [8,26]. For example, CsI has long been used as an effective scintillator because Cs and I possess considerable stopping powers for the rays [27]. As a result of its broadband gap, the de-excitation process occurs with the emission of UVC light in the 3.7- to 4.3-eV energy region [28]. After adding the dopant Na, a VIS blue light with an energy of about 3 eV is emitted from the CsI:Na film [6]. Hsu et al. investigated the luminous properties of pure CsI and CsI:Na films prepared by thermal vacuum evaporation under different substrates and post-annealing temperatures [6]. The 310 nm luminescence arising from the electron-hole recombination via self-trapped excitons (STEs) is characteristic of the pure CsI film. Its doping Na induced the broad-banded 400–450 nm luminescence originated from other electron-hole recombinations involving newly created energy states [3]. Other dopants, such as thallium (Tl), europium (Eu), and cerium (Ce), are also applicable [26,29].

The decay time, defined as the time after which the intensity of the light decreases to 1/e of its maximum, is important for scintillators in fast counting/timing applications [30]. Without doping, the decay time is usually speedy, e.g., 1 ns and 10 ns orders for pure CsI [31] and NaI [32]. The decay time is extended due to doping when generating additional energy states in VIS light. For example, the decay time of CsI:Na and CsI:Tl is about 460 and 1000 ns, respectively [33,34]. Compared to doped CsI, doped NaI possesses a much shorter decay time, e.g., about 230 ns for NaI:Tl [35] which enables NaI-based crystals to be suitable scintillators with fast responses. In 1948, the discovery of NaI:Tl as an inorganic scintillating compound resulted in the first practical photomultiplier tube (PMT) for a gamma-ray spectrometer [36]. Shepherd reported the fabrication and characterization of thin NaI:Tl films of up to 130 cm^2^ areas by using standard bulk load evaporation (BLE) and powder flash evaporation (PFE) [37]. It was found that PFE films exhibited higher light yields than BLE ones, and these yields could be predicted by controlling the substrate temperature and the composition of the source [37]. The skull method, a simple and cost-effective technique commonly used for casting metals, was applied to the growth of NaI:Tl crystals [38]. The results indicated that, even under minimal thermal convection, the smooth distribution of Tl led to the minimal spread of the scintillation characteristics across the ingot [38]. NaI:Tl-based scintillators can be used as detectors in various fields, such as medical imaging, automobiles, radioisotope identification devices, and homeland security [39,40,41,42]. For example, Kim et al. developed a miniature scintillation camera using a NaI:Tl scintillator and a position-sensitive photomultiplier tube (PSPMT) for clinically imaging malignant breast tumors [39]. Siciliano et al. presented experimental data and computer simulations of NaI:Tl-based gamma-ray detectors in radiation portal monitor (RPM) systems for homeland security applications at international borders [40].

Although Tl is an efficient dopant of NaI, it is highly toxic and can lead to Tl poisoning upon skin exposure. Another disadvantage of NaI:Tl-based scintillators is that they are easily affected by environmental fluctuations such as temperature and electronic noise, which may cause peak shift phenomena in measuring scattering spectra [43]. Therefore, it is interesting to search for another dopant in NaI. Na-doped CsI, one of the brightest available scintillators, has emission wavelengths between 280 and 350 nm [6]. CsI:Na crystals have been studied in detail with a variety of fabrication methods, such as direct seeding [44], thermal vacuum deposition [25], and the edge-defined film-fed growth (EFG) method [45].

In this study, Cs was doped into the NaI thin film used for the scintillator. Mainly, that is because NaI is more hygroscopic than CsI and is susceptible to moisture. Therefore, hermetic encapsulation of a scintillator is usually required for long-term stability by coated hydrophobic protective layers to prevent moisture [42,46]. Examples of these protective materials include SiO_2_, Parylene N [6], and the composition of fluoroplastic varnish and ethyl acetate [47]. This study reports the fabrication and characterization of Cs-doped NaI (NaI:Cs) films. These films were grown by thermal vacuum co-evaporation, and protective Parylene-N layers were deposited on them. The crystallinities of the films were evaluated by the X-ray diffraction (XRD) patterns, and the luminescence efficiencies under UVC exposure were investigated.

## 2. Materials and Methods

NaI and NaI:Cs films are grown on B270 and fused silica substrates with a thickness of about 1 mm and a diameter of 1 inch, using thermal resistance evaporation in a vacuum environment. B270 glass slides were first cleaned with alcohol in an ultrasonic bath, blow-dried with nitrogen gas, and placed in a substrate holder about 25 cm above the molybdenum (Mo) boat. Next, the approximately 15 × 20 × 70 mm boat was filled with pure NaI or its mixture with CsI powder for co-evaporation. The total weight of the powder was about 50 g, with 1 wt.% CsI for NaI:Cs(1%) sample and 2 wt.% for NaI:Cs(2%) sample. Then, a diffusion pump evacuated the vacuum chamber to a 2 × 10^−5^ torr base pressure. Then, the Mo boat was preheated with an electric current of 50 A to remove the moisture from the NaI powder in the outgassing process, increasing the pressure to 2~5 × 10^−4^ torr. The outgassing process was completed until the vacuum pressure was again decreased to the near base pressure. Then, adjusting the electric current to about 200 A and maintaining the substrate temperature at 200 °C, NaI, NaI:Cs(1%), or NaI:Cs(2%) materials located within the Mo boat were entirely deposited onto the substrate of about 10 µm thickness at a rate of approximately 25 nm/s by a quartz monitor during deposition with a rotational substrate holder.

Since NaI is hygroscopic and susceptible to environmental moisture, a protective Parylene-N layer was subsequently deposited onto the samples when the substrate temperature naturally decreased below 120 °C [48,49,50]. Parylene N, or poly-p-xylylene, possesses the advantages of low gas/water permeability, surface hydrophobicity, and transparency at wavelengths above 400 nm. The 5 g Parylene-N dimer was sublimated by a ceramic evaporation boat and was pyrolyzed into the monomer during thermal radiation from the perforated and heated Mo foil. During the thermal decomposition procession, the sample surface was finally deposited with the polymer layer of the Parylene N at about 800 nm thickness under a deposition rate of ~0.7 nm/sec.

The as-deposited NaI samples have a protective Parylene-N layer to prevent moisture; however, the moisture can somewhat wet the NaI sample in the air. Therefore, dehumidification in a vacuum condition was needed for the NaI samples to renew the photoluminescence property and to prevent chemical reactions with water. During dehumidification, NaI samples were annealed at a rate of 5 °C/min and kept at 120 °C for 20 min in a vacuum pressure of ~2 × 10^−5^ torr.

The NaI powder and sample XRD patterns were obtained using an X-ray diffractometer Rigaku Multiflex (Rigaku Co., Spring, TX, USA). The luminescence properties were investigated by using a homemade system mainly consisting of a UVC LED (LG Innotek Co., Jung-gu, Seoul, Republic of Korea) and a spectrometer (USB2000, Ocean Optics, Inc., Largo, FL, USA), as illustrated in Figure 1a. Figure 1b shows the spectrum of this light source, indicating a maximum emission intensity of about 273 nm and its FWHM of 12.0 nm. Driven by a forward voltage of 8 VDC, the LED source could provide a flux of 10 mW. Moreover, to maintain its working temperature below 60 °C, it was cooled via a thermoelectric cooling module equipped with a CPU fan. The photoluminescence light emitted by the Cs-doped NaI samples was collected by a UV collimator lens, and the spectrum ranging from 200 to 850 nm was analyzed using the spectrometer USB2000 (Ocean Optics, Inc., Largo, FL, USA).

## 3. Results and Discussion

### 3.1. XRD Characteristic of Pure NaI and NaI Cs-Dopped Films

The measured XRD pattern of the NaI powder used in the deposition process is shown in Figure 2. The main complex orientations of (111), (002), (022), (113), (222), (004), (133), (024), (224), and (115) are indicated after comparison to the reference XRD database from the Materials Project data (https://next-gen.materialsproject.org/) accessed on 6 October 2023.

#### 3.1.1. XRD Characteristic of NaI Films on the Glass Substrate

Figure 3a shows the XRD patterns of the as-deposited undoped NaI, NaI:Cs(1%), and NaI:Cs(2%) films grown on B270 glass substrates. Compared to pure NaI, some additional peaks appear due to Cs-doping, especially the (004) one. As the Cs-doping concentration increases to 2%, some peaks, including (024) and (115), disappear. That is, the Cs dopant changes the NaI polycrystalline structure.

For the post-annealed samples, their XRD patterns are illustrated in Figure 3b. The crystalline orientations of NaI films with 1-wt.% and 2-wt.% Cs-dopants are primarily along the (022), (113), (024), and (115) planes. Compared to the undoped film, the Cs-doping inhibits the (002) and (133) peaks as well as accentuating the (002) and (113) peaks. In addition, the effects of post-annealing on the crystalline orientations vary for different doping concentrations. For the NaI:Cs(1%) film, this annealing eliminates many peaks, including the obvious (002) and (004) ones. Moreover, for the NaI:Cs(2%) film, this process makes the (002) peak disappear but brings the (024) one out.

#### 3.1.2. XRD Characteristic of NaI Films on Fused Silica Substrate

For samples deposited on fused silica substrates, the as-deposited XRD patterns are shown in Figure 4a. Compared to that grown on a B270 slide, the pure NaI film exhibits extra (111) and (224) peaks, while the obvious (002) one disappears. As a result of 1% Cs-doping, the (111) peak vanishes, but the (002) one appears and dominates. This dominant peak suddenly disappears as the Cs-doping concentration increases to 2%.

Figure 4b shows the XRD patterns of the post-annealed samples. For both undoped and Cs-doped NaI films, common orientations along (022), (113), (024), and (115) are observed. This post-annealing process eliminates the (111) and (224) peaks for the pure NaI sample, removes the dominant (002) peak for the NaI:Cs(1%) sample, and retains all peaks for the NaI:Cs(2%) sample. The post-annealing process tended to orient these films in certain directions to a stable structure.

#### 3.1.3. XRD Characteristic Variation under Different Deposition Treatment

It is worth noting that peak splitting, where two peaks are too close to be distinguished based on their orientations, is observed for all post-annealed samples. Li et al. found that when annealed at 800 °C, the (Ge:SiO_2_)/SiO_2_ multilayers exhibited an apparent splitting (fine structure) of the Ge (220) XRD peak and possessed a preferred orientation [51]. It was suggested that this phenomenon arose from compressive stress exerted on Ge nanocrystals, causing an orthorhombic distortion of the diamond structure of bulk Ge [51]. The XRD peak splitting in post-annealing was also found in materials such as ZnGa_2_O_4_, W_18_O_49_, and CH_3_NH_3_PbI_3_ films [52,53,54]. One of the possible explanations is that the post-annealing process causes phase decomposition/segregation in the sample. The heating leads to a strain variation along the depth. Down to a certain thickness, the layer grows coherently to the buffer zone, and any further growth partially relaxes the structure [55,56]. That was why the as-deposited NaI and NaI:Cs(1%) samples had the peak-splitting phenomenon, and the NaI:Cs(2%) sample had no sensation. Moreover, the post-annealing heat changed the NaI:Cs(2%) sample to peak-splitting.

This is the first study to characterize the crystalline orientations of NaI films. Table 1 lists the XRD peaks of pure NaI, NaI:Cs(1%), and NaI:Cs(2%) films grown on B270 and fused silica substrates before and after annealing.

Many studies investigating the XRD patterns of pure CsI and doped CsI films deposited by thermal vacuum evaporation exhibited different crystal structures depending on various conditions, including film thickness, deposition rate, substrate, and post-annealing temperatures. In our previous study, multiple peaks appeared at (110), (200), (211), (220), and (310) for pure CsI films deposited at Ts = 200 °C, while only two primary peaks showed at (200) and (310) for the 0.5-wt.% and 1-wt.% Na-doped films. The (200) peak disappeared, but the smaller peaks (110) and (211) appeared as the Na-doping concentration increased to 2 wt.% [6]. Lebedynskiy et al. studied CsI films containing 0.09 wt.% of Eu deposited with Ts = 573 K exhibiting a polycrystalline structure without growth texture. Increasing Ts and reducing wt.% of Eu resulted in the appearance of predominant orientations along the (100), (211), (220), and (330) planes [57]. Cha et al. studied the 1.0 mol% Tl-doped CsI scintillating layers fabricated by thermal evaporation at Ts = 30 and 200 °C without rapid thermal annealing; they had multiple XRD peaks at (110), (200), (211), (220), (310), (222), (321), and (400), which were almost equal to those of cubic phase in pure CsI film. However, the primary peaks at (200) and (220) were sparingly observed at post-annealing at 250 °C for 30 min [58]. All these findings suggest that substrate heating and/or post-annealing tend to rearrange the doped CsI films toward only one or few crystalline orientations. Furthermore, these films with only one or a few preferred orientations were shown to provide better luminescence performance than those with several crystal planes [59].

In the present study, the substrate temperature maintained at 200 °C ensures that the fabricated NaI and NaI:Cs films possess columnar structures. Post-annealing in the vacuum significantly also reduces the number of XRD diffraction peaks. The film’s structure is improved and may benefit the photoluminescence performance. The micro-columns perpendicular to the substrate have the properties of high light collimation, decrease the lateral spreading of scintillating light, and increase the resolution in UV/radiation detection [54,55]. Based on the Thornton zone model, the structure and topography of thick sputtered or thermally evaporated coatings depend on the deposition conditions, including working pressure, melting temperature (Tm) of the material, and substrate temperature (Ts). The columnar structures are formed when Ts/Tm increases above 0.5 but below 0.75 [48,49]. In this study, a Ts/Tm of 0.51 (Tm = 933 K for NaI) and deposition at low vacuum pressure resulted in the formation of columnar structures.

However, NaI is a very deliquescence material that reacts significantly with moisture in the air. The surface or cross-section’s topography is unlike the CsI film studied in SEM in our previous study because the structure becomes disorganized when the NaI film is removed from the Parylene-N preventative layer before SEM measurement. This leads to the need to evaluate the crystallite size of the NaI film. Crystallite size generally corresponds to the coherent volume of the material for the respective diffraction peak. Sometimes, it also fits with the size of the grains of a sample or the thickness of polycrystalline thin film or bulk material. Unfortunately, the size, usually assessed with Scheller’s equations, is also meaningless because complex XRD patterns have different crystal sizes in the NaI film.

### 3.2. Photoluminescence of NaI and NaI:Cs Films

This study used a UV light source to induce NaI and NaI:Cs films PL. Figure 5 shows a schematic diagram of the PL energy band structure corresponding to an undoped NaI film and a NaI:Cs film irradiated with UV light. Valence band (VB) electrons absorb 4.54 eV photon energy from radiated UV light at a wavelength of 273 nm and excite from path A1 to the exciton level. However, photon energies below the 5.8 eV NaI band gap cause electrons to fail to reach the conduction band (CB) and STEs directly below CB. As shown in its XRD plot, the grown NaI film is polycrystalline and may produce additional exciton levels due to inherent defects created during deposition [60]. Thus, UV light may excite electrons from path A2 to the CB level. The electrons then decay to the STE level and relax to the activator ground states. Then, the PL light (E1) emits a wavelength of about 3.955 eV, as shown in Figure 6.

The electrons in exciton levels may decay to activator excited states and defect levels. Then, they relax to activator ground states, the PLs of E2, E3, and E4 with peak energies around 3.955, 3.047, and 2.853 eV, respectively. The photo energies of the emitted lights are described by E4 > E2 > E3, as shown in Figure 5. E4 is identical to E2 in the same activator excited state, where the energy gap may be wide due to the complex polycrystalline shown in the XRD pattern of Figure 3 and Figure 4. The primary peak wavelength is located at the PL of E2. The less amount of the E4 excitons due to the slightly higher energy state and the less amount of the E3 excitons to transfer the defect levels cause less luminescent intensities than E4, as shown in Figure 5 and Figure 6.

#### 3.2.1. Photoluminescence of NaI Films on a Glass Substrate

Figure 6a shows the NaI:Cs(1%) sample has the higher luminescent intensity because Cs dopants can fill the detects, as mentioned above [61]. Additional activator (Cs) excited states and activator ground states may be generated in the band gap, leading to the photon energy transfer from the host NaI to the dopant Cs of the excited state [62]. As previously mentioned, such a process tends to rearrange pure or doped NaI films toward only one or a few crystalline orientations, shown in Figure 4a, in turn enhancing their PL performance compared to those with more orientations.

Leblans et al. prepared CsI:Eu needle crystal layers by physical vapor deposition and investigated their luminescence properties before and after thermal annealing [63]. The results indicated that the luminescence intensity of as-deposited layers was low, and annealing (170 °C in the air for one hour) enhanced photo- and radio-luminescence on average by a factor of 1.5~3 [63]. The emission spectra of CsI:Pb were studied under different annealing temperatures from 220 to 370 °C, showing that their relative intensities depended strongly on this temperature [64]. For example, the emission of the aggregates was primarily observed at low temperatures, and a slight low-energy shift in most emission bands occurred with the increasing temperature [64]. In addition to alkali metal halides, the annealing process was also reported to improve the luminescent efficiency in other types of scintillators [65,66,67].

Figure 6 compares the PL spectra of pure NaI film and NaI:Cs film samples deposited on B270 substrates (a) before and (b) after annealed treatments with the same set of samples. The intensities of the post-annealed samples are almost the same. As mentioned above, the small plot in Figure 6b illustrates that Cs-doped NaI films have better luminescence properties. However, the annealing process only increases the peak intensity at 3.047 eV by about 10%. Since the optimal structure of the NaI film can be easily obtained during the deposition process, the efficiency of annealing to improve luminescence is reduced.

#### 3.2.2. Photoluminescence of NaI Films on Fused Silica Substrate

The PL spectra of the as-deposited and post-annealed samples grown on fused silica substrates are shown in Figure 7. Two prominent peaks are also observed. The intensity of the first peak of the NaI film is similar to that of the B270 substrate; however, the wavelength at ~285 nm (4.32 eV) is less than that at ~314 nm (3.955 eV) on the B270 substrate. The intensity of ~407 nm (3.047 eV) is much smaller than that of the B270 substrate. The as-deposited NaI:Cs(1%) film has the most significant intensity, and pure NaI performs better than NaI:Cs(2%) samples, as shown in Figure 7a. It is noted that the luminescence efficiency does not always increase with increasing doping concentration. A general trend indicates that optimal doping enhances the performance of PL.

The luminescence efficiencies of pure NaI, NaI:Cs(1%), and NaI:Cs(2%) increase by about 1.7, 1.5, and 4.3 times, respectively, due to the post-annealing process. However, as shown in Figure 7b, the peak intensity of all three post-annealed samples increases to approximately 260 (a.u.), four times smaller than the sample deposited on the B270 glass substrate. In addition, this post-annealing does not affect the 4.32 eV peak intensity.

An intense blue emission at around 3.955 eV and a weaker peak at about 4.32 eV are observed. The 4.32 eV fluorescence arises from a small number of electrons in the excited state before irradiation. After exposure to UVC, they have enough energy to be excited to the CB and then emit light through the intrinsic STE [68], as shown in the schematic diagram E1 in Figure 5. The intensity at 3.047 eV increases by means of impurities in alkali metal halides, such as NaI and CsI, generating additional energy states in the forbidden gap, enhancing their PL performance. For example, Balamurugan et al. grew pure and Tl-doped CsI single crystals by the melt growth (Bridgman) technique. They found that for a doping concentration of 0.06 mol%, the emission intensity increased by six-fold c [66] compared to the undoped crystal [69]. The influence of Tl doping on the PL of CsI:Tl polycrystalline layers was investigated, showing that the peak intensity in the spectrum increased (from 0 to 1100 ppm of doped Tl) and then saturated (from 1100 to 2400 ppm of doped Tl) with the Tl concentration [70]. Also, the peak intensity of 310 nm is the highest in CsI:Na(1 wt.%) in our previous studies [6] and 306 nm NaI:Cs (1%) in this study.

### 3.3. PL Affected by Fused Silica and B270 Substrate

Fused silica and B270 glass substrates are used in this study. Although the incident UVC light source is not associated with luminescence during doping or annealing processes, as shown in Figure 6 and Figure 7, the transmittance of fused silica substrates under UV LED light illumination at 273 nm peak is usually more significant than that of B270 substrates. However, under the same UVC irradiation, the luminous efficiency of NaI films deposited on B270 glass substrates is more effective than that of those on fused silica substrates.

Fused silica is a high-purity form of silicon dioxide (SiO_2_) glass without any formed crystalline structure. The absence of a regular crystal lattice in fused silica gives it unique characteristics. However, B270 glass is a specialized material developed and manufactured by the German company SCHOTT AG. It is primarily composed of borosilicate, including silicon dioxide (SiO_2_), boron oxide (B_2_O_3_), aluminum oxide (Al_2_O_3_), sodium oxide (Na_2_O), and potassium oxide (K_2_O). The adhesion of alkali metal halide film deposited on fused silica substrate is inferior to that of the B270 substrate in our previous studies, which also appears to be the luminescence efficiency in this study. The B270 borosilicate glass with impurity oxides is superior to fused silica in the field of luminescence for the deposited NaI films on the interfaces. Improvements in the post-annealing process of NaI polycrystalline films are suitable for deposition on B270, but not all on fused silica due to the splitting and decreasing of the XRD peaks in Figure 4b and the luminescent limitation in Figure 7b. As the results mentioned above, XRD patterns and luminescence characteristics are sensitive to the substrate interface.

The 120 °C temperature and vacuum condition in the post-annealing process are designed to prevent moisture oxidation in the NaI film because the luminescent efficiency decreases with moisture in the NaI film and the oxidation of the NaI film. Furthermore, the post-annealing process can renew the NaI film to the optimal luminous state, though moisture has penetrated the film.

## 4. Conclusions

This study first prepared pure NaI and Cs-doped NaI films by thermal vacuum evaporation. The Parylene-N layer was sequentially deposited on the NaI film to prevent it from moisture. Their XRD and luminescence properties investigated the deposited NaI films. The optimal photo-luminescent intensities can increase by co-evaporation of 1 wt.% CsI and by post-annealing at 120 °C for 20 min under a vacuum of ~2 × 10^−5^ torr. When a 273 nm (4.54 eV) UV LED illuminates the approximately 10 µm deposited NaI film, two major luminescence peaks in the spectrum are observed: 3.047 eV light from intrinsically defective and/or activator excited states and 3.955/4.32 eV light from intrinsic STE based on fused silica/B270 substrates. Moreover, the film deposited on the B270 glass substrate was more than four times more luminescent than that deposited on the fused silica substrate. It can be observed that the X-ray diffraction pattern and luminescence characteristics are sensitive to the interface of the substrate. In addition, the post-annealing process at 120 °C under vacuum conditions can restore the NaI films to their optimal luminous state. These findings will undoubtedly contribute to developing optimal scintillator films for high-energy photo-imaging technology.

## Figures and Tables

**Figure 1 nanomaterials-13-02747-f001:**
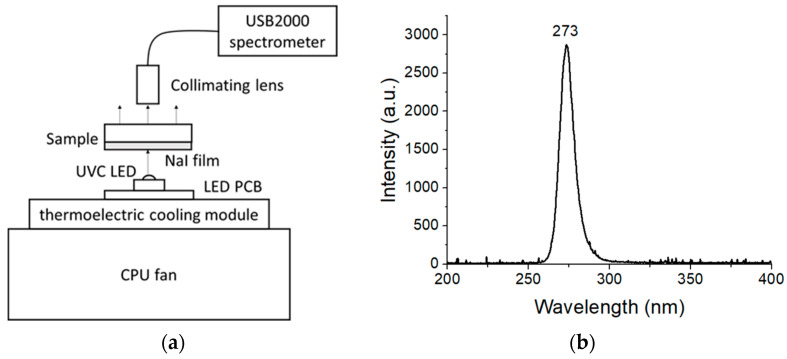
(**a**) Schematic diagram of the homemade photoluminescence system. (**b**) The maximum emission intensity in the UV LED light source spectrum is about 273 nm, and the FWHM is 12.0 nm.

**Figure 2 nanomaterials-13-02747-f002:**
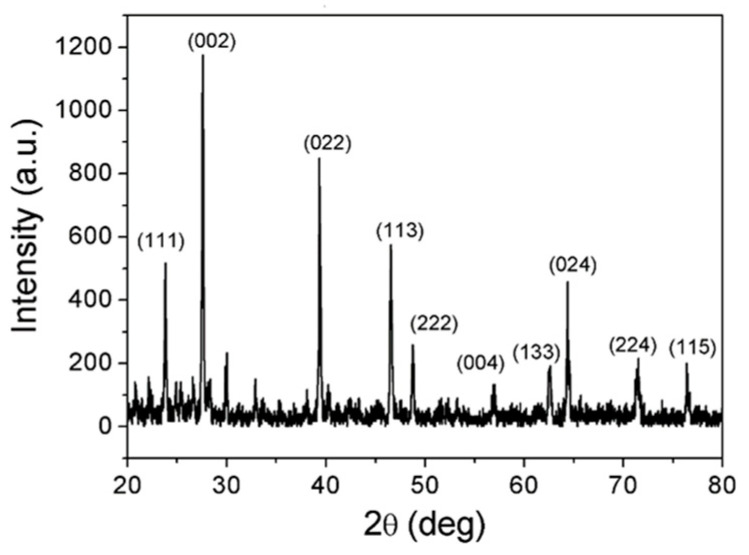
The measured XRD pattern of the NaI powder used in the deposition process.

**Figure 3 nanomaterials-13-02747-f003:**
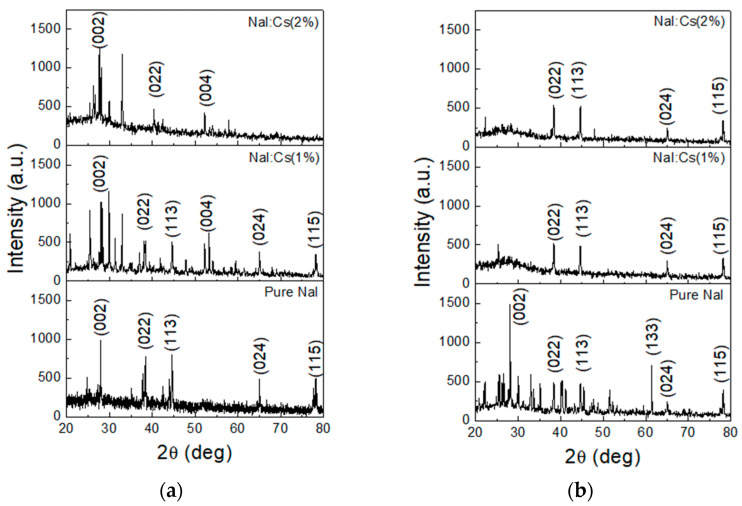
XRD patterns of (**a**) the as-deposited and (**b**) the post-annealing undoped NaI, NaI:Cs(1%), and NaI:Cs(2%) films grown on the B270 substrate.

**Figure 4 nanomaterials-13-02747-f004:**
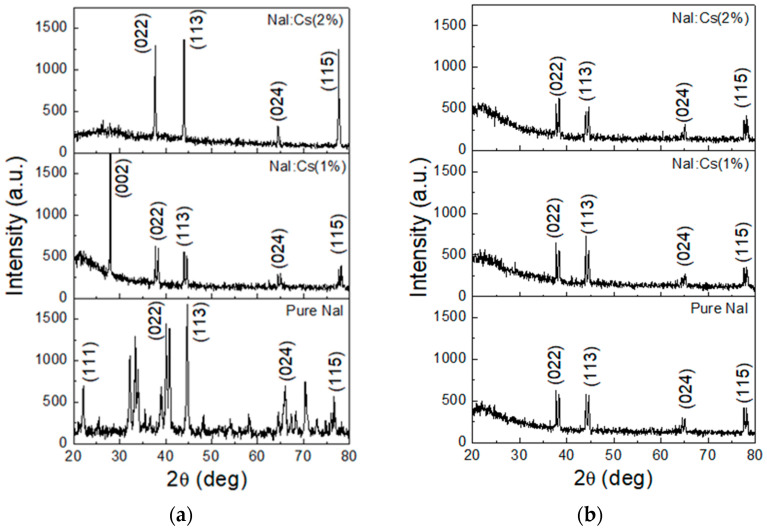
XRD patterns of the (**a**) as-deposited and (**b**) post-annealing undoped NaI, NaI:Cs(1%), and NaI:Cs(2%) films grown on fused silica substrates.

**Figure 5 nanomaterials-13-02747-f005:**
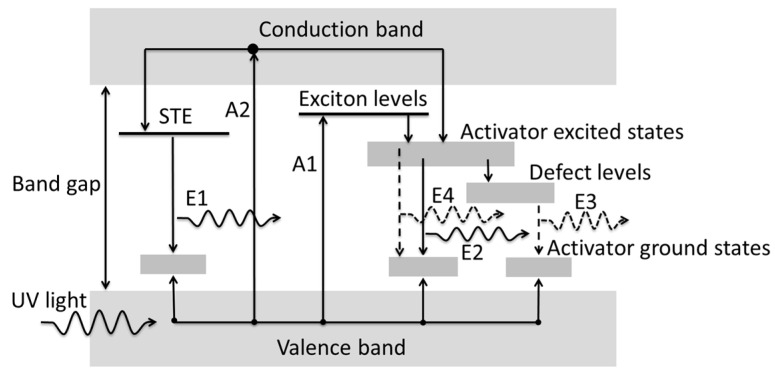
Schematic diagram of PL energy band structures corresponding to NaI films irradiated with UV light.

**Figure 6 nanomaterials-13-02747-f006:**
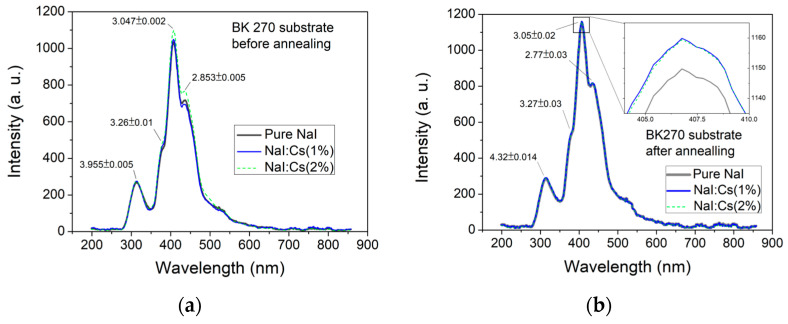
PL spectra under 273 nm excitation light of pure NaI, NaI:Cs(1%), and NaI:Cs(2%) films grown on B270 substrates (**a**) before and (**b**) after annealing treatments with the same set of samples.

**Figure 7 nanomaterials-13-02747-f007:**
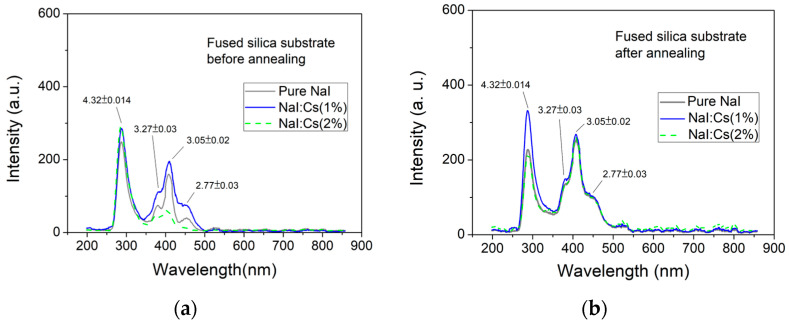
PL spectra of undoped NaI (0%), NaI:Cs(1%), and NaI:Cs(2%) films grown on fused silica substrates (**a**) before and (**b**) after annealing.

**Table 1 nanomaterials-13-02747-t001:** XRD peaks of pure NaI, NaI:Cs(1%), and NaI:Cs(2%) films grown under different conditions.

Sample	Substrate	Annealing	Dominant Peak(s)	Other Observable Peak	Peak Splitting
NaI powder	-	before	(002)	(111), (002), (022), (113), (222), (004), (133), (024), (224), (115)	No
pure NaI	B270	before	(002)	(022), (113), (024), (115),	(022), (113), (115)
		after	(002)	(022), (113), (133), (024), (115)	(113)
NaI:Cs(1%)	B270	before	(002)	(022), (113), (024), (004), (115)	(002), (022)
		after	(022), (113)	(024), (115)	No
NaI:Cs(2%)	B270	before	(002)	(022), (004)	(002)
		after	(022), (113)	(024), (115)	No
pure NaI	Fused silica	before	(022), (113)	(111), (024), (224), (115),	(022)
		after	(022), (113)	(024), (115)	All
NaI:Cs(1%)	Fused silica	before	(002)	(022), (113), (024), (115),	(022), (113)
		after	(022), (113)	(024), (115)	All
NaI:Cs(2%)	Fused silica	before	(022), (113)	(024), (115)	All
		after	(022), (113)	(024), (115)	All

## Data Availability

The data are included in the article.

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
