# Peer review of "Photoluminescence of Cesium-Doped Sodium Iodide Films Irradiated by UV LED"

_nanomaterials, 2023, doi:10.3390/nano13202747_

Round 1

Reviewer 1 Report

Dear Editor,

Thanks for inviting me for reviewing the manuscript entitled Photoluminescence of cesium-doped sodium iodide films irradiated by UV LED authored by Hsing-Yu Wu and Jin-Cherng Hsu, et al.

The author studied the annealed Cs-doped NaI film scintillators with enhanced luminescence performance. The authors discussed the crystal structure changes before and after Cs doping and post annealing and argued the as induced luminescence. The manuscript may be accepted after addressing the following issues.

1)      “This post-annealing process eliminates the (111) and (224) peaks for the pure NaI 191 sample, removes the dominant (002) peak for NaI:Cs(1%) sample, and retains all peaks for 192 NaI:Cs(2%) sample.” Please further explain why they disappear.

2)      Please estimate the light output for sample films.

3)      What are the decay times of Cs-doped samples at room temperature and ambient pressure?

4)      Some references are not well formatted.

The English language is good for understanding the manuscript.

Author Response

Response to Comments and Suggestions

Thanks for inviting me for reviewing the manuscript entitled Photoluminescence of cesium-doped sodium iodide films irradiated by UV LED authored by Hsing-Yu Wu and Jin-Cherng Hsu, et al.

The author studied the annealed Cs-doped NaI film scintillators with enhanced luminescence performance. The authors discussed the crystal structure changes before and after Cs doping and post annealing and argued the as induced luminescence. The manuscript may be accepted after addressing the following issues.

Response:

The authors thank this reviewer for all the helpful comments.

1) “This post-annealing process eliminates the (111) and (224) peaks for the pure NaI 191 sample, removes the dominant (002) peak for NaI:Cs(1%) sample, and retains all peaks for 192 NaI:Cs(2%) sample.” Please further explain why they disappear.

Response:

These are what we have observed. The only thing we can say about these is that “the post-annealing process tended to orient these films along certain directions to a stable structure.” We have added this sentence to the revised manuscript (Line 194).

2) Please estimate the light output for sample films.

Response:

This study is the first property study of NaI:Cs films. The thickness of the film should be known for estimating the light output. However, sodium iodide (NaI) and cesium iodide (CsI) are very hygroscopic. The structures of the NaI and NaI:CsI films are very easily destroyed by moisture. The cross-section SEM was not performed to measure the precise thicknesses of the films. Therefore, the light output cannot be precisely and quantitatively estimated.

3) What are the decay times of Cs-doped samples at room temperature and ambient pressure?

Response:

This manuscript focuses on the crystal structures and photoluminescence phenomena of NaI and NaI:Cs samples. We didn’t measure/calculate the decay time of these films. We will be doing this in the near future.

4) Some references are not well formatted.

Response:

We have revised these references.

Reviewer 2 Report

Paper:

 Photoluminescence of cesium-doped sodium iodide films irradiated by UV LED

The authors investigated the structures and luminescent properties of undoped sodium iodide (NaI) and cesium-doped NaI (NaI:Cs) films deposited by thermal vacuum evaporation.

They offered XRD characteristic of pure NaI and NaI Cs-dopped films, variation under different deposition treatment, Photoluminescence spectra of NaI and NaI:Cs Films and comparitions with other author results,

What is missing to this paper is a SEM or a imaging test result for one of the probe, as in other papers.

Also I recommend to cite few additional papers:

https://doi.org/10.3390/bios11120497  - as example of SEM co-investigation for applications.

https://doi.org/10.3390/nano12132141 - for formamidinium lead halide perovskite nanocrystals high X-ray attenuation coefficient and bright luminescence.

Author Response

Response to Comments and Suggestions

The authors investigated the structures and luminescent properties of undoped sodium iodide (NaI) and cesium-doped NaI (NaI:Cs) films deposited by thermal vacuum evaporation.

They offered XRD characteristic of pure NaI and NaI Cs-dopped films, variation under different deposition treatment, Photoluminescence spectra of NaI and NaI:Cs Films and comparitions with other author results,

Response:

The authors thank this reviewer for all the helpful comments.

What is missing to this paper is a SEM or a imaging test result for one of the probe, as in other papers.

Response: Based on the Thornton zone model, the structure and topography of thick sputtered or thermally evaporated coatings depend on the deposition conditions, including working pressure, melting temperature (Tm) of the material, and substrate temperature (Ts). The columnar structures are formed when Ts/Tm increases above 0.5 but below 0.75. In this study, a Ts/Tm of 0.51 (Tm = 933 K for NaI) and deposition at low vacuum pressure resulted in the formation of columnar structures. The cross-section image is more important than the surface image because the excellent cylindrical structure of the cross-section image illustrates that the film has significant photo-luminescence and light-guiding properties in our previous studies of CsI film. In this study, we have tried to measure the cross-section of the NaI film. However, the NaI film is more hygroscopic than the CsI film. Once we crosscut the NaI sample, it absorbed water from the moisture and became a spherical structure, not a cylindrical CsI film structure in our previous study. The moisture destroys the image, so we didn’t display the SEM image. For example, in our previous study “Luminescence of CsI and CsI:Na Films under LED and X-ray Excitation,” only 2 wt% NaI dopant can make the aggregate structure due to moisture penetrating. Not to mention that this film has the main NaI material formed in this study.

Also I recommend to cite few additional papers:

https://doi.org/10.3390/bios11120497 - as example of SEM co-investigation for applications.

https://doi.org/10.3390/nano12132141 - for formamidinium lead halide perovskite nanocrystals high X-ray attenuation coefficient and bright luminescence.

Response:

We didn’t cite these references because we think they are different samples and it is not fair to make a comparison among them.

Reviewer 3 Report

This is a good article that should be recommended for publication, however some comments should be taken in to account.

1.     It is not clear from the introduction why this article was sent to this journal “Nanomaterials”

2.     Line 29-30.  Unclear sentence.  1.  “A scintillator is a photo-luminescent material”. It must be simply “is a luminescent material”.   2. Line 30. Here is not only X-ray and gamma rays must be mentioned, but also electron, ion and neutron particles!!! See, some references below:

Altman, M. R., Dietrich, H. B., Murray, R. B., & Rock, T. J. (1973). Scintillation response of NaI (Tl) and KI (Tl) to channeled ions. Physical Review B7(5), 1743.

Popov, A. I., Chernov, S. A., & Trinkler, L. E. (1997). Time-resolved luminescence of CsI- Tl crystals excited by pulsed electron beam. Nuclear Instruments and Methods in Physics Research Section B: Beam Interactions with Materials and Atoms122(3), 602-605.

Shen, F., Fu, Q., Huang, T., & Wang, W. (2022). A compact dual gamma neutron detector based on NaI (Tl+ Li) scintillator readout with SiPM. Crystals12(8), 1077.

3.     Paragraph 3.2.   Clarify what activator levels are we talking about here? If activator Tl is meant, then it is necessary to consider direct excitation a of the activator in one of its own absorption bands (A, B, C and D).

4.     Fig.6. Please clarify the wavelength of the excitation light.

5.     Fig.6 and 7. In order to identify individual luminescence bands, it is necessary to decompose the luminescence spectra into Gaussian components, and this should be done in energy units on the horizontal axis (eV!!!)

6.     In the conclusions, it is necessary to formulate more clearly what new data on the studied materials were obtained in this work?

In general, the manuscript is interesting and can be recommended for publication after constructive reflection on the above comments.

Author Response

Response to Comments and Suggestions

This is a good article that should be recommended for publication, however some comments should be taken in to account.

Response: The authors thank this reviewer for all the helpful comments.

  1. It is not clear from the introduction why this article was sent to this journal “Nanomaterials”

Response: We used the “Journal Finder” tool in MDPI website to find the most suitable journal for submission. “Nanomaterials” came out to be the first one in the Match Rank.

  1. Line 29-30. Unclear sentence. 1. “A scintillator is a photo-luminescent material”. It must be simply “is a luminescent material”. 2. Line 30. Here is not only X-ray and gamma rays must be mentioned, but also electron, ion and neutron particles!!! See, some references below:

Altman, M. R., Dietrich, H. B., Murray, R. B., & Rock, T. J. (1973). Scintillation response of NaI (Tl) and KI (Tl) to channeled ions. Physical Review B, 7(5), 1743.

Popov, A. I., Chernov, S. A., & Trinkler, L. E. (1997). Time-resolved luminescence of CsI- Tl crystals excited by pulsed electron beam. Nuclear Instruments and Methods in Physics Research Section B: Beam Interactions with Materials and Atoms, 122(3), 602-605.

Shen, F., Fu, Q., Huang, T., & Wang, W. (2022). A compact dual gamma neutron detector based on NaI (Tl+ Li) scintillator readout with SiPM. Crystals, 12(8), 1077.

Response: We have modified these sentences and added relative references.

  1. Paragraph 3.2. Clarify what activator levels are we talking about here? If activator Tl is meant, then it is necessary to consider direct excitation a of the activator in one of its own absorption bands (A, B, C and D).

Response: The authors are not sure about what the meanings of Tl, A, B, C and D are. The energy band structure shown in Figure 5 is used to explain the spectra in Figure 6 (the gray peaks at 312, 308, 406, and 430 nm).

As described in Sec. 3.2, Valence band (VB) electrons absorb 4.54 eV photon energy from radiated UV light at a wavelength of 273 nm and excite from path A1 to the exciton level. However, photon energies below the 5.8 eV NaI band gap cause electrons to fail to reach the conduction band (CB) and STEs directly below CB. As shown in its XRD plot, the grown NaI film is polycrystalline and may produce additional exciton levels due to inherent defects created during deposition [57]. Thus, UV light may excite electrons from path A2 to the CB level. The electrons then decay to the STE level and relax to the Activator ground states. Then, the PL light (E1) emits a wavelength of about 312 nm, as shown in Figure 6.

The electrons in Exciton levels may decay to Activator excited states and Defect levels. Then they relax to Activator ground states, the PLs of E2, E3, and E4 with peak wavelengths around 380, 406, and 430 nm, respectively. The photo energies of the emitted lights are described by E4 > E2 > E3, as shown in Figure 5. E4 is identical to E2 in the same Activator excited state, where the energy gap may be wide due to the complex polycrystalline shown in the XRD pattern of Figures 3 and 4. The primary peak wavelength is located at the PL of E2. The less amount of the E4 excitons due to the slightly higher energy state and the less amount of the E3 excitons to transfer the defect levels cause less luminescent intensities than E4, as shown in Figures 5 and 6. 

  1. Fig.6. Please clarify the wavelength of the excitation light.

Response: The spectrum of the excitation light is shown in Figure 1(b), indicating a maximum emission intensity at about 273 nm. We have added this to the revised “Figure 6. PL spectrum under 273 nm excitation light ….” in Line 307.

  1. Fig.6 and 7. In order to identify individual luminescence bands, it is necessary to decompose the luminescence spectra into Gaussian components, and this should be done in energy units on the horizontal axis (eV!!!)

Response: The Reviewer gives us a good suggestion. The luminescence spectra should be a Gaussian component in the eV horizontal axis. However, the main decomposed peak of the spectra clearly shows three peaks and a shoulder. We cancel the gray part in Figures 6 & 7 and show the peak position (eV) and its rms value using eV during the short time of revising the manuscript.

  1. In the conclusions, it is necessary to formulate more clearly what new data on the studied materials were obtained in this work?

Response: We have revised and added a few more sentences to address this in the revision.

In general, the manuscript is interesting and can be recommended for publication after constructive reflection on the above comments.

Reviewer 4 Report

NaI thin film with Cs dopant was prepared and the luminescent properties were studied. The thin films were examined with XRD. UVC light was used to excite the materials and a homemade photoluminescence system was made to record the luminescent signals. There are several major issues with the current manuscript.

1. The comparison of the luminescent intensity is quite arbitrary without mentioning how to keep the thickness (or the evenness of the film) and the absorbance of the thin film consistency, as these factors could swing the intensities of luminescent. Moreover, it is recommended to repeat each experiment three times and report the error bars for these measurements.

2. Can the authors comment on the type of luminescent, is this fluorescent or phosphorescent, and what is the lifetime of emission?

3.  The authors may want to discuss how glass substrate and fused silica substrate absorb in the UVC region, and how this would affect the measurement. Thus, the conclusion that the film deposited on the glass substrate was more luminescent than the fused silica substrate needs more justification.

4. CsI thin film with Na dopant was prepared by similar methods and showed induced luminescence, the authors need to explain the need to prepare NaI thin film with Cs dopant. More importantly, why select Cs as the dopant?  

Author Response

Response to Comments and Suggestions

NaI thin film with Cs dopant was prepared and the luminescent properties were studied. The thin films were examined with XRD. UVC light was used to excite the materials and a homemade photoluminescence system was made to record the luminescent signals. There are several major issues with the current manuscript.

  1. The comparison of the luminescent intensity is quite arbitrary without mentioning how to keep the thickness (or the evenness of the film) and the absorbance of the thin film consistency, as these factors could swing the intensities of luminescent. Moreover, it is recommended to repeat each experiment three times and report the error bars for these measurements.

Response: The thickness of all samples was controlled to be around 10 µm “by a quartz monitor during deposition with a rotational substrate holder.” (added in Line 132) The deposited film can be uniform for small substrates as small as 1 inch in diameter (added in Line 119).  And the weight of the deposited NaI powder was kept at 50g in each deposition. However, the structures of the NaI and NaI:CsI films are very easily destroyed by moisture. The cross-section SEM was not performed to measure the precise thicknesses of the films for the absorbance of the thin film consistency.

We focus on qualitatively analyzing the effects of Cs-doping, post-annealing, and different substrates on these samples' crystal structures and photoluminescence phenomena. We cancel the gray part in Figures 6 & 7 and show their peak positions and their rms values using eV from the average results of the several experimental samples for the repeat experiment. We will try to measure the precise thickness to obtain quantitative data such as the decay time and light output in the near future.

  1. Can the authors comment on the type of luminescent, is this fluorescent or phosphorescent, and what is the lifetime of emission?

Response: This manuscript focuses on the crystal structures and photoluminescence phenomena of NaI and NaI:Cs samples. We didn’t measure/calculate the decay time of these films. We will be doing this in the near future. But, the type of luminescence is fluorescent light because those of luminescence CsI and NaI materials all belong to fluorescent light. 

  1. The authors may want to discuss how glass substrate and fused silica substrate absorb in the UVC region, and how this would affect the measurement. Thus, the conclusion that the film deposited on the glass substrate was more luminescent than the fused silica substrate needs more justification.

Response: We have discussed this in Section 3.3, Line 349. Fused silica is a high-purity form of silicon dioxide glass without forming any crystalline structure. The absence of a regular crystal lattice in fused silica gives it unique characteristics. However, B270 glass is primarily composed of borosilicate, including silicon dioxide, boron oxide, aluminum oxide, sodium oxide, and potassium oxide. The B 270 with impurity oxides is superior to fused silica in luminescence for the deposited NaI films on the interfaces. “XRD patterns and luminescence characteristics are sensitive from the substrate interface.” That is added on Line 361.

  1. CsI thin film with Na dopant was prepared by similar methods and showed induced luminescence, the authors need to explain the need to prepare NaI thin film with Cs dopant. More importantly, why select Cs as the dopant?

Response: The reasons why NaI and NaI:Cs films were prepared and investigated are detailed in the manuscript. This is the first study of the Cs-deposited NaI films to investigate their XRD patterns and luminescence properties.

As described in the Introduction, compared to doped CsI, doped NaI possesses a much shorter decay time, e.g., about 230 ns for NaI:Tl. That enables NaI-based crystals to be suitable scintillators with fast responses. Also, Although Tl is an efficient dopant of NaI, it is highly toxic and can lead to Tl poisoning upon skin exposure. Another disadvantage of NaI:Tl-based scintillators is that they are easily affected by environmental fluctuations such as temperature and electronic noise, which may cause peak-shift phenomena in measuring scattering spectra. Therefore, it is interesting to search for another dopant in NaI.

Round 2

Reviewer 1 Report

The authors addressed part of issues that this reviewer raised. Below are a following feedback to author's response.

1) Line 191, it should be (022) not (002) right? please double check.

2) for author's response point 3, "This manuscript focuses on the crystal structures and photoluminescence phenomena of NaI and NaI:Cs samples. We didn’t measure/calculate the decay time of these films. We will be doing this in the near future." 

The authors focused on structure of NaI and Cs-NaI, however, the XRD of pure NaI is wired, please provide the reference XRD for pure NaI, or powder diffraction file (PDF) for the NaI crystal.

3) The references were not revised, there are still not of unformatted references (e.g., 1, 2, 5, 6, 8, 10.....).

References of research articles should involve authors, title, year, issue, pages. Many cited references have not pages.

I am not qualified to assess the quality of English in this paper

Author Response

Response to Reviewer 1 Comments and Suggestions

The authors addressed part of issues that this reviewer raised. Below are a following feedback to author's response.

Thank you for pointing this out. We agree with this comment. Therefore, we have revised below:

1) Line 191, it should be (022) not (002) right? please double check.

Response: We have changed (002) in Line 191 to (022).

2) for author's response point 3, "This manuscript focuses on the crystal structures and photoluminescence phenomena of NaI and NaI:Cs samples. We didn’t measure/calculate the decay time of these films. We will be doing this in the near future."

The authors focused on structure of NaI and Cs-NaI, however, the XRD of pure NaI is wired, please provide the reference XRD for pure NaI, or powder diffraction file (PDF) for the NaI crystal.

Response: The measured XRD pattern was compared to the reference shown in the attached figure (from the Materials Project data, https://next-gen.materialsproject.org/materials/mp-23268?formula=NaI#diffraction_patterns) to identify all peaks. Due to the copyright issue, we didn’t include this figure in the manuscript. We have modified the manuscript as follows:

“The NaI powder and sample…” in Line 148.

“The measured XRD pattern…” in Line 163.

“…are indicated after comparison to the reference XRD database from the Materials Project data (https://next-gen.materialsproject.org/).” In Line 165~166.

Figure 2.: “The measured” XRD pattern of the NaI powder used in the deposition process” in Line 168.

3) The references were not revised, there are still not of unformatted references (e.g., 1, 2, 5, 6, 8, 10.....).

References of research articles should involve authors, title, year, issue, pages. Many cited references have not pages.

Response: We have checked and modified the references.

Reviewer 3 Report

After successful revision, this manuscript can be recommended for publication.

There is only a need to correct chemical formula in references [3, 4, 5]

Author Response

Response to Reviewer 3 Comments and Suggestions  

After successful revision, this manuscript can be recommended for publication.

There is only a need to correct chemical formula in references [3, 4, 5]

Response: Thank you for pointing this out. We agree with this comment. Therefore, we have revised below:

We have modified these references.

Reviewer 4 Report

The biggest concern raised by the reviewer is not addressed. Without knowing the sample thickness (around 10 um is not enough as the changes were only 10%) and repeating the measurements, the comparison of emission intensity shouldn't lead to a conclusive result.    

Author Response

Response to Reviewer 4 Comments and Suggestions

Response: Thank you for pointing this out. We agree with this comment. Therefore, we have revised below:

The finding that the annealing process increased the peak intensity at 3.047 eV by about 10% is not affected by the thickness of the sample. The same sample was measured before and after annealing, so the thickness was fixed. In other words, Figure 6(b) was obtained with the same set of samples in Figure 6(a).

We have modified the following sentence in Line 302~304: Figure 6 compares the PL spectra of pure NaI film and NaI:Cs film samples deposited on B270 substrates (a) before and (b) after annealed treatments with the same set of samples.

The thickness of all samples was controlled to be around 10 µm by a quartz monitor during deposition with a rotational substrate holder. For small substrates as small as 1 inch in diameter, the deposited film can be uniform. Also, the weight of the deposited NaI powder were kept at 50g in each deposition.

The attached figures are the SEM images of the NaI films, which were destroyed by moisture and changed the columnal structure to a spherical aggregation. The thickness was measured to be about 10 um

Round 3

Reviewer 4 Report

most concerns are addressed.